# Disulfide disruption reverses mucus dysfunction in allergic airway disease

Leslie E. Morgan[1,13], Ana M. Jaramillo[1,13], Siddharth K. Shenoy [2,3], Dorota Raclawska[1], Nkechinyere A. Emezienna[1,4], Vanessa L. Richardson[1], Naoko Hara[1], Anna Q. Harder[1], James C. NeeDell[1], Corinne E. Hennessy[1], Hassan M. El-Batal[5], Chelsea M. Magin[1,5], Diane E. Grove Villalon[6], Gregg Duncan [2,7], Justin S. Hanes[2,3,8,9], Jung Soo Suk [2,3,9], David J. Thornton [10], Fernando Holguin[1], William J. Janssen[1,11,12], William R. Thelin[6] & Christopher M. Evans [1,12✉]

Airway mucus is essential for lung defense, but excessive mucus in asthma obstructs airflow, leading to severe and potentially fatal outcomes. Current asthma treatments have minimal effects on mucus, and the lack of therapeutic options stems from a poor understanding of mucus function and dysfunction at a molecular level and in vivo. Biophysical properties of mucus are controlled by mucin glycoproteins that polymerize covalently via disulfide bonds. Once secreted, mucin glycopolymers can aggregate, form plugs, and block airflow. Here we show that reducing mucin disulfide bonds disrupts mucus in human asthmatics and reverses pathological effects of mucus hypersecretion in a mouse allergic asthma model. In mice, inhaled mucolytic treatment loosens mucus mesh, enhances mucociliary clearance, and abolishes airway hyperreactivity (AHR) to the bronchoprovocative agent methacholine. AHR reversal is directly related to reduced mucus plugging. These findings establish grounds for developing treatments to inhibit effects of mucus hypersecretion in asthma.

[1] Department of Medicine, School of Medicine, University of Colorado, Aurora, CO, USA. [2] Center for Nanomedicine at the Wilmer Eye Institute, Johns Hopkins University School of Medicine, Baltimore, MD, USA. [3] Department of Ophthalmology, Johns Hopkins University School of Medicine, Baltimore, MD, USA. [4] Department of Obstetrics and Gynecology, Howard University College of Medicine, Washington, DC, USA. [5] Department of Bioengineering, College of Engineering, Design, and Computing, University of Colorado, Denver | Anschutz Medial Campus, Denver, CO, USA. [6] Parion Sciences, Inc., Durham, NC, USA. [7] Fischell Department of Bioengineering, School of Engineering University of Maryland, College Park, MD, USA. [8] Department of Pharmacology & Molecular Sciences, Johns Hopkins University School of Medicine, Baltimore, MD, USA. [9] Department of Chemical & Biomolecular Engineering, Johns Hopkins University, Baltimore, MD, USA. [10] Wellcome Trust Centre for Cell-Matrix Research and the Lydia Becker Institute of Immunology and Inflammation, School of Biological Sciences, The University of Manchester, Manchester, UK. [11] Department of Medicine National Jewish Health, Denver, CO, USA. [12] Department of Immunology and Microbiology, School of Medicine, University of Colorado, Aurora, CO, USA. [13]These authors contributed equally: Leslie E. Morgan, Ana M. Jaramillo. ✉email: Christopher.Evans@cuanschutz.edu

With daily exposures to >8000 liters of air containing billions of particles and potential pathogens, respiratory tissues embody the need for robust host defense. Airway mucus is critical for protection, but poor control of mucus function is central to numerous lung diseases[1–5]. In patients who die during asthma exacerbations, mucus obstruction is a feature long-recognized by pathologists[6], with plugging observed in >90% of cases and thus considered a major cause of fatal obstruction[5]. Mucus obstruction is also prominent in non-fatal cases of severe asthma[2], but effective mucolytic therapies are lacking. Accordingly, asthma treatments could be significantly improved by determining molecular mechanisms of mucus dysfunction[7].

The predominant macromolecules in mucus are polymeric mucin glycoproteins[7,8]. In health, effective host defense requires homeostatic mucin synthesis and secretion[1]. By contrast, excessive mucin production and secretion are demonstrated by alcian blue-periodic acid Schiff's (AB-PAS) staining within airway surface and submucosal gland epithelial cells, and in plugs filling small airways in fatal asthma (Fig. 1a). Mucin glycoprotein overproduction is also prominent in mild-to-moderate disease[9], suggesting a potentially broad etiological role for mucus hypersecretion in asthma.

Two polymeric mucin glycoproteins are abundant in airway mucus. The chief airway mucin in healthy mucus is MUC5B, whose absence in mice results in impaired mucociliary clearance, pathogen accumulation, and spontaneous lethal infections[1]. In humans with asthma, MUC5B decreases in many patients[9,10], and lower MUC5B levels are associated with worsened disease[10]. On the other hand, excessive MUC5B is also a risk factor for developing pulmonary fibrosis[11,12], and *Muc5b* has a gene dosage effect on lung fibrosis in mice[13]. The other airway polymeric mucin, MUC5AC, is expressed at low levels at baseline, but it is dramatically up-regulated in human asthma[9,10] and in mouse models[14] where it is required for asthma-like mucus obstruction and airway hyperreactivity (AHR)[15].

Taken together, MUC5AC and MUC5B play significant functions in the lungs, but their precise roles in health and disease are complex[7]. The genetic, developmental, and environmental factors that cause aberrant *MUC5AC* and *MUC5B* gene expression are active areas of investigation. However, given the inherent heterogeneity among these pathways, finding selective gene expression or signal transduction targets that can prevent mucus dysfunction while still preserving (or improving) mucus defense remains an on-going challenge[7,8,16]. As an alternative, we propose that pathological effects of mucus hypersecretion can be reversed while healthy functions are enhanced by directly targeting mucin glycoprotein polymers.

Polymeric mucins are evolutionary precursors of the hemostasis protein von Willebrand factor (vWF). Accordingly, they possess vWF-like amino (N-) and carboxyl (C-) terminal cysteine-rich domains that form covalent disulfide intermolecular linkages to form large glycopolymers[17]. MUC5AC and MUC5B first assemble in the endoplasmic reticulum as C-terminal disulfide dimers. Then in the Golgi, along with becoming heavily glycosylated, they further multimerize via N-terminal disulfide linkages[18]. Upon secretion, MUC5AC and MUC5B become hydrated, extend into strands, and form a porous mesh-like gel that traps particles and mediates mucociliary clearance.

Due to the nature of mucus as a gel polymer, it is exquisitely sensitive to changes in the concentrations of solid materials within its matrix. Accordingly, when mucins are overproduced and hypersecreted in asthma, mucociliary clearance dysfunction reflects aberrant gel behavior that can be corrected by disrupting its polymeric mucin components[8]. Therefore, here we test whether aberrant mucus gel structure, mucociliary clearance, and

airway obstruction in asthma can be improved by disrupting mucin disulfides with the reducing agent tris(2-carboxyethyl) phosphine (TCEP) (Fig. 1b).

We chose TCEP due to its stability in aqueous solutions compared to reducing agents such as dithiothreitol, and also due to the ability of TCEP to rapidly reduce disulfides, including mucins[19,20]. In a recent study using a mouse model of chronic mucus dehydration similar to CF, TCEP was able to disrupt mucus plugs in chronically obstructed airways[21]. Accordingly, we hypothesized that TCEP could be applied to test the ability of disulfide reduction to reverse the effects of acute mucus hypersecretion in a setting of allergic asthma.

We show that TCEP disrupts human asthmatic mucus and reverses pathological effects of mucus hypersecretion in a mouse allergic asthma model. These effects of TCEP were mediated by loosening of mucus gel structure, enhancement of mucociliary clearance, and reduction of airway obstruction. These data support the therapeutic potential of mucolytic treatments to inhibit the effects of mucus hypersecretion in muco-obstructive diseases, including asthma.

## Results

**Polymeric mucins are targets for disulfide disruption of mucus in human asthma.** Using mucus from human asthma patients, we verified the presence of targets that could be reduced with TCEP under physiologic conditions (pH 7.4, 37 °C). In histologic samples from fatal asthma, AB-PAS positive plugs were sensitive to TCEP as demonstrated by alkylation of reduced thiols with biotinylated maleimide (Fig. 1c). Furthermore, in mucus aspirated from bronchial airways of the same patient, TCEP reduced the sizes of massive MUC5AC and MUC5B polymers in a concentration dependent manner (Fig. 1d). Based on these findings, we tested whether mucin polymer disruption could improve the physical properties of asthmatic mucus.

We used multiple particle tracking to assess mucus biophysical properties by quantifying mean square displacement (MSD) of 2 μm diameter carboxylated micro-particles. In fatal asthma mucus, MSD was significantly increased in samples treated with TCEP (Fig. 1e, f). When converted to rheologic parameters[22,23], these results demonstrated that TCEP treatment rendered mucus less viscoelastic (Supplementary Fig. 1a), an effect that was driven by a significant reduction in its elastic modulus (Supplementary Fig. 1b). The rapid depolymerization of mucins and improvement of rheologic properties in asthmatic mucus in vitro suggested that TCEP could improve mucus functions in an animal model of asthma in vivo.

**Mucolytic treatment reverses mucus dysfunction in a mouse model of allergic asthma.** To produce allergic asthma-like inflammatory, mucous, and AHR phenotypes, BALB/c mice were exposed to a fungal allergen, *Aspergillus oryzae* extract (AOE), by aerosol weekly for a total of four challenges[15]. Endpoints were studied 48 h after the last AOE challenge, a time point of robust inflammation and mucin overproduction (Supplementary Fig. 2a)[15]. Nebulized TCEP was used to determine the effects of mucolytic treatment on mucus properties and functions in vivo.

To evaluate mucus directly on mouse airway surfaces, we employed multiple particle tracking to quantify MSD of muco-inert nanoparticles (MIPs) aerosolized onto mouse tracheas ex vivo[24–26]. Tracheas were removed from saline or AOE challenged mice, opened, placed on glass coverslips, and treated with nebulized 100 nm diameter MIPs suspended in saline vehicle or TCEP (Fig. 2a). Compared to uninflamed controls, MIP diffusion was heterogeneous and impaired in the tracheal mucus of AOE-challenged mice (Fig. 2b), resulting in a 2.7-fold decrease ($p = 0.02$) in median MSD values (Fig. 2c). Upon mucolytic

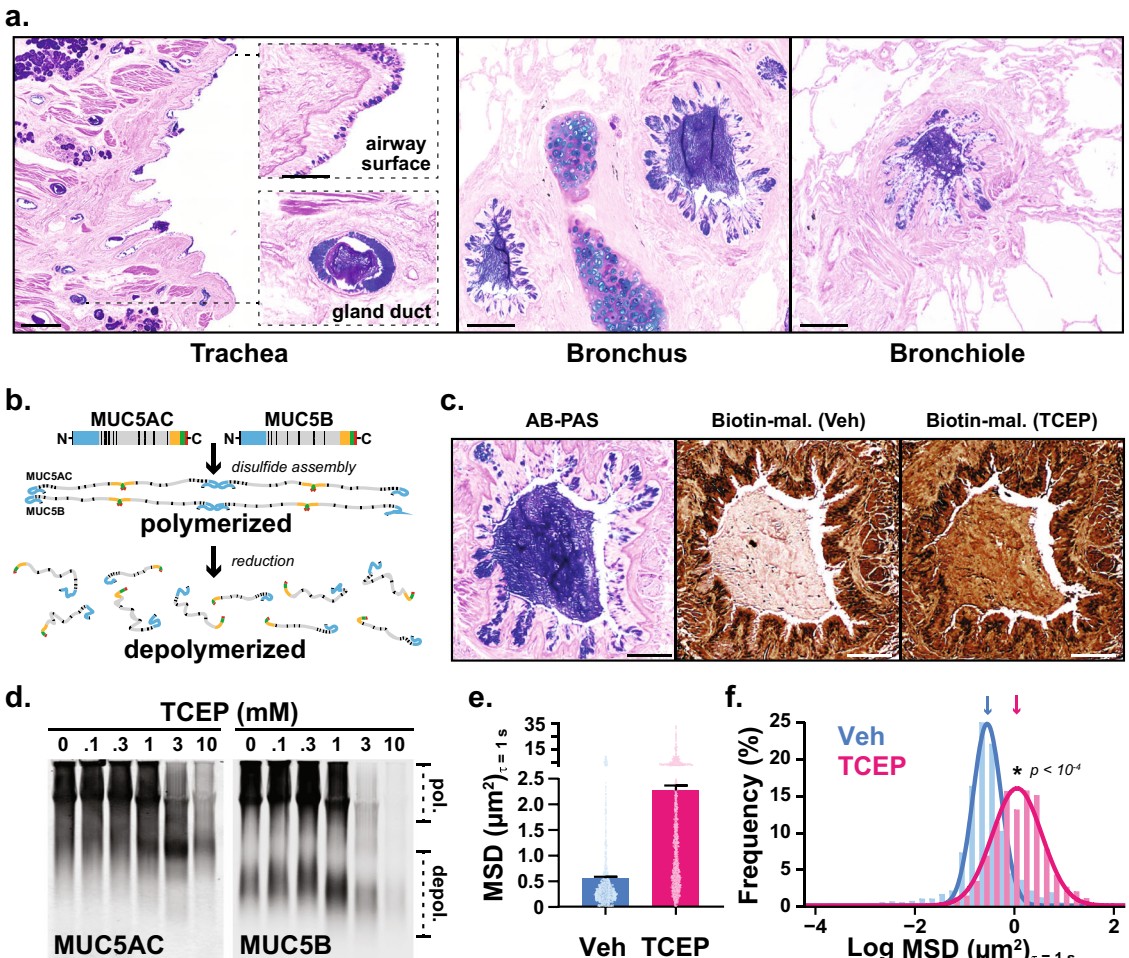

**Fig. 1 Polymeric mucins in asthmatic airways are targets for disulfide disruption. a** Alcian blue/periodic acid-Schiff (AB-PAS) stained tissues from the lungs of patients ($n = 2$) who died during asthma exacerbations demonstrate mucin glycoproteins in large and small airways. Scale bars, 500 μm and 100 μm in insets (trachea), 200 μm (bronchus and bronchiole). **b** MUC5AC and MUC5B assemble via amino (N-) and carboxyl (C-) terminal disulfide bonds that are sensitive to reducing agents. **c** A mucus plug in a fatal asthma airway was examined in consecutive sections stained with AB-PAS (purple), or labeled with biotinylated-maleimide (brown) after incubation at 37 °C for 10 min with saline vehicle (Veh) or tris(2-carboxyethyl)phosphine (TCEP, 10 mM). Scale bars, 250 μm; image representative of airways from $n = 2$ patients. **d–f**. Expectorated sputum ($n = 3$ patients) was treated with TCEP (0.1–10 mM, 37 °C, 30 min), separated by electrophoresis (1% SDS/agarose, non-reducing), and detected by immunoblot MUC5AC and MUC5B (**d**). Note that full reduction causes epitope loss for both anti-mucin antibodies resulting in decreased signal intensities at higher TCEP concentrations in **d**. Diffusion of 2-μm carboxylated micro-particles was evaluated in mucus samples aspirated from fatal asthma bronchi (**e**, **f**). Compared to vehicle controls (Veh, cyan, $n = 1303$), particle mean square displacement (MSD) increased significantly after TCEP (magenta, $n = 1425$). Bars in **e** are means ± sem of summarized linear-scale values, with individual points (open circles). Data in **f** are log distributions of particles with curves showing Gaussian non-linear fits ($r^2 = 0.99$ for Veh and TCEP). Significance was determined by two-tailed Mann–Whitney U-tests with p-value and "*" denoting significance from Veh in **f**. Inverted arrows in **f** indicate locations of median values. Source data are provided as a Source Data file.

treatment, MSD normalized to levels and homogeneity that were indistinguishable from non-allergic controls (Fig. 2b, c), reflecting an increase in computed mucus mesh spacing (Supplementary Fig. 3). Taken together, these findings suggested that mucolytic treatment normalized mucus gel microstructure uniformity, which we postulated would also improve mucociliary function.

We thus investigated the efficacy of inhaled mucolytic treatment on mucociliary clearance in allergically inflamed mice in vivo. Animals were allergen challenged with AOE, and 48 h after the challenge, they received nose-only aerosol treatments with TCEP (5–500 mM) for 40 min (see Supplementary Fig. 2b). Immediately after mucolytic treatment, lungs were lavaged, and the disruption of mucin polymers and the elimination of inflammatory cells from airspaces were assessed[13]. In TCEP-treated mice, mucins demonstrated faster electrophoretic migration relative to controls thereby validating effective depolymerization

(Fig. 2d and Supplementary Fig. 4). Effects of TCEP on mucolysis and mucociliary clearance were dose dependent, with mucins demonstrating significant depolymerization in mice treated with aerosols of 50 and 500 mM TCEP solutions (Fig. 2e, f and Supplementary Fig. 5). Concordant with mucolysis, there were acute decreases in inflammatory cell numbers in lung lavage fluid (Fig. 2f and Supplementary Table 1). With increasing doses of TCEP, eosinophils were cleared resulting in a >5-fold decrease in total numbers recovered from lung lavage from mice treated with the highest concentration of TCEP delivered (500 mM). The rapid elimination of leukocytes from airway surfaces upon mucolytic treatment provided direct functional evidence of improved clearance[13].

Collectively, our in vitro, ex vivo, and in vivo results showed that mucolytic treatment reversed mucus dysfunction in a setting of allergic inflammation and excessive mucin production. These

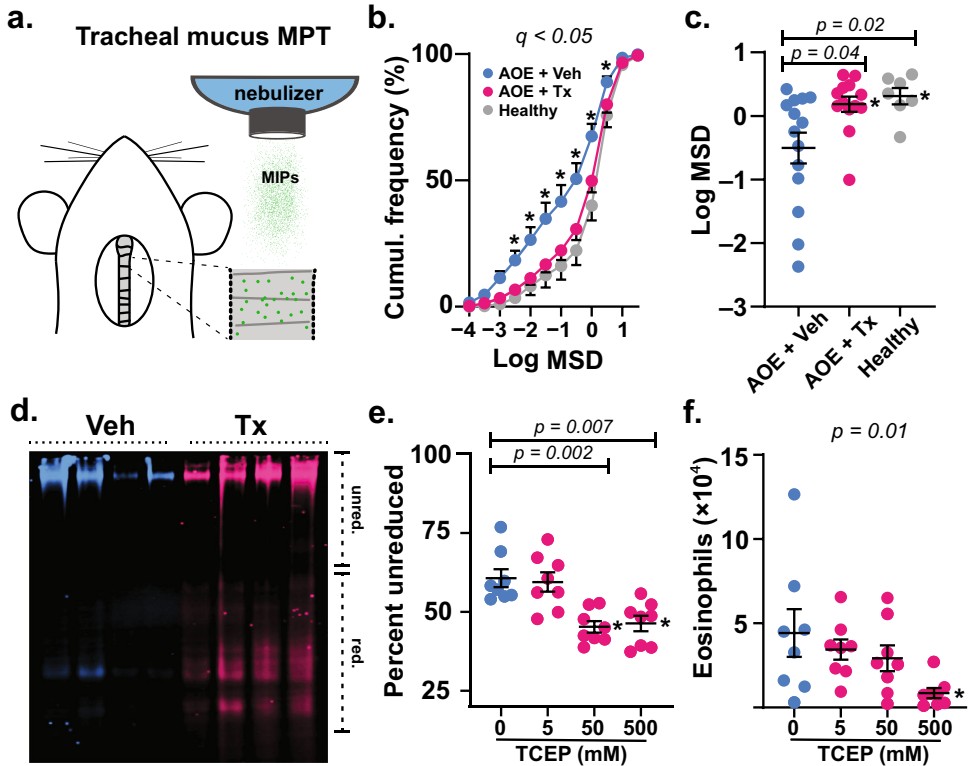

**Fig. 2 Mucolytic treatment improves mucus function in allergic mouse airways. a** Mucus was probed in tracheal preparations ex vivo using 100 nm muco-inert particles (MIPs). **b**, **c** In *Aspergillus oryzae* extract (AOE) challenged mice receiving 500 mM aerosol TCEP treatment (Tx, magenta, $n = 13$ biological replicates), MIP diffusion measured as mean square displacement (MSD) increased significantly compared to vehicle (Veh, cyan, $n = 14$ biological replicates) and non-AOE exposed animals (Healthy, gray, $n = 7$ biological replicates). Cumulative distribution data in **b** show the distributions of MSD values (means ± sem). Scatter plot data in **c** show individual median MSD values per animal. **d**–**f** Mucociliary clearance was tested in AOE challenged mice treated by nose-only aerosol with TCEP in a concentration dependent manner or Veh, followed by immediate lung lavage. Lectin blot analysis using *Ulex europaeus* agglutinin I (UEA-1) (α1,2-fucose) in **d** shows disruption of mucin polymers in TCEP-treated animals (Tx, 500 mM, magenta) compared to vehicle (cyan). Image shows four samples per group. Intensities of high molecular weight (polymerized) and low molecular weight (depolymerized) mucins were evaluated using Image Studio software (**e**). Numbers of total eosinophils (**f**) recovered in lung lavage decreased significantly in Tx (magenta, $n = 8$ biological replicates) vs. Veh (cyan circles, $n = 6$ biological replicates) exposed animals. Lines and error bars in **c**–**f** are means ± sem. '*' denotes significance using a cut-off of 0.05 determined by unpaired two-tailed $t$-tests using a two-stage step-up method at a 5% false discovery rate from Veh in **b**, by two-tailed Mann–Whitney $U$-test in **c**, and by non-parametric one-way ANOVA in **e**, **f**. Source data are provided as a Source Data file.

findings led us to hypothesize that mucolytic treatment could also protect against asthma-like airflow obstruction. To test this, we developed a two-route challenge and treatment analysis, with methacholine administered intravenously (i.v.) to cause obstruction, and with TCEP administered by inhalation to cause mucolysis. Methacholine and TCEP treatments were given during pulmonary function tests (Supplementary Fig. 2c).

Since we previously found that Muc5ac absence prevents AHR to inhaled methacholine, which results in both airway smooth muscle contraction and mucin secretion[15], we first tested whether Muc5ac was similarly required for AHR to methacholine administered intravenously (without mucolytic treatment). Compared to non-allergic animals, AOE challenged mice demonstrated significantly exaggerated increases in total lung resistance ($R_L$) and airway resistance ($R_{AW}$) in response to i.v. methacholine (Fig. 3a–d). Importantly, these AHR responses were abolished in *Muc5ac* gene deficient mice (Fig. 3a–d), thereby validating a role for mucus hypersecretion in this methacholine challenge model. The degrees of protection observed in this chronic mucus prevention setting also established benchmarks for comparing effects of acute mucolytic rescue. Accordingly, we next tested whether inhaled TCEP protected mice from AHR by reversing mucus plugging.

In AOE exposed wild type mice treated with inhaled TCEP during i.v. methacholine challenge, we observed significant

improvements in $R_L$ and $R_{AW}$ (Fig. 3a–d). There was also significant protection of resistance ($G_{TI}$) and elastance ($H_{TI}$) in peripheral lung tissues (Fig. 3e–h). Thus, inhaled TCEP was not acutely detrimental to pulmonary surfactant function, and it appeared to confer protection from AHR by preserving airway patency. Indeed, the protective effects of TCEP treatment in allergic wild type mice were indistinguishable from benchmarks in allergic *Muc5ac*$^{-/-}$ mice ($p = 0.98$), strongly supporting the hypothesis that mucolytic protection from AHR was linked to reversal of mucus plugging.

To validate the role of mucolysis in protection from AHR in TCEP-treated mice, we confirmed mucin polymer reduction by examining lung lavage fluid via immunoblot (Supplementary Fig. 6). We also quantified mucin accumulation in airspaces histologically[15]. During methacholine-induced bronchoconstriction, secreted mucin volume, airway obstruction, and heterogeneous plugging were significantly reversed throughout the airways of TCEP-treated animals (Fig. 4). Thus, protection from AHR was directly related to the ability of mucolytic treatment to reduce airway plugging.

## Discussion
The studies reported here show that disrupting mucin polymers improves mucus microstructure, enhances mucus transport, and

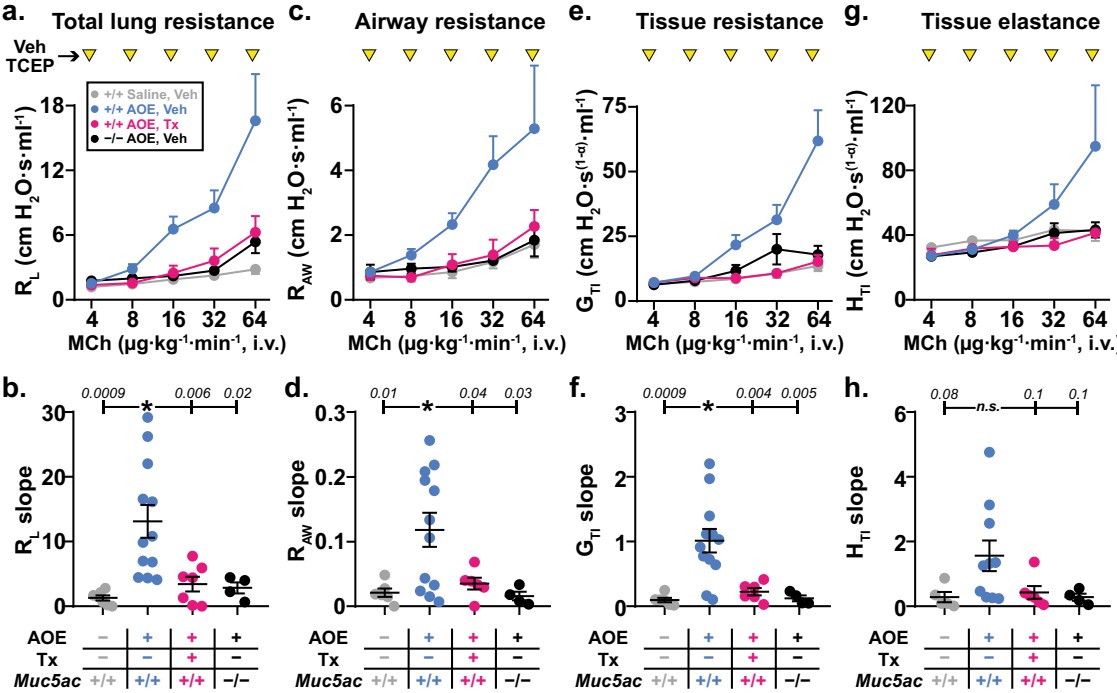

**Fig. 3 Mucolytic treatment reverses allergic airway hyperreactivity.** Dose response curves to methacholine (MCh, 4–64 µg kg$^{-1}$ min$^{-1}$, i.v.) were generated in AOE challenged allergic wild type (WT) mice (magenta, $n = 7$ biological replicates) treated with TCEP between MCh doses (100 mM, inverted yellow triangles). For comparison, saline challenged non-allergic WT mice (gray, $n = 6$ biological replicates), AOE challenged allergic WT mice (cyan, $n = 12$ biological replicates), and AOE challenged $Muc5ac^{-/-}$ mice (black, $n = 4$ biological replicates) were treated with nebulized vehicle (Veh) during i.v. MCh dose response tests. Values are means ± sem. For each mouse, dose response curves were fitted by log-linear best-fit regression analysis, and slopes of regression lines were analyzed by one-way ANOVA. "*", $p < 0.05$ using Dunnett's test for multiple comparisons relative to AOE-challenged Veh-treated WT mice (p-values are shown). Total lung resistance ($R_L$ in **a**, **b**), conducting airway resistance ($R_{AW}$ in **c**, **d**), tissue resistance ($G_{TI}$ in **e**, **f**) and tissue elastance ($H_{TI}$ in **g**, **h**) are shown. Source data are provided as a Source Data file.

protects airflow in allergic asthma settings. In patients, mucin overproduction and pathologic changes in mucus biophysical properties are correlated with asthma exacerbations and fatalities[2,5,7,9,27]. These features are observed frequently by pathologists, but they are usually ignored during clinical assessments since diagnostic tests and interventions have lagged. On the diagnostic side, mucus obstruction is becoming more recognizable through high resolution imaging[2], but treatment options are still constrained by a lack of efficacious mucolytics.

Indeed, while potentially beneficial in laboratory settings, mucolytics and expectorants are not widely used therapeutically. The only FDA-approved reducing agent available as an inhaled mucolytic is N-acetylcysteine (NAC), but its efficacy is low due to NAC's weak activity at airway pH and high mucin concentrations[28–30]. We propose that inhaled mucolytic agents can reduce plugging and thus improve airway function in asthma. Nonetheless, it will be critical for any strategy that reverses the detrimental effects of mucus hypersecretion to do so while also protecting mucus-mediated defense. Since disulfide assembly is a critical process in the formation of viscoelastic mucus, mucin reduction remains an attractive target for disrupting mucus plugs in human airways (see Fig. 1). In addition, it is also plausible that the use of a reducing agent could affect other targets. For example, oxidants released from cells as a result of injury and inflammation could be directly neutralized, but the effects of transient antioxidant treatment need to be tested[13].

In contrast to the findings reported here, a recent study investigating mucociliary transport showed inhibitory effects of TCEP on mucus transport in non-diseased pigs[31]. However, that investigation focused on tracheobronchial glandular secretions,

and it was conducted using large (500–600 µm diameter) metal particles. Given the loads imparted by these materials and the forces required to displace them, inhibitory consequences of TCEP treatment on exogenous transport in that report may not be directly comparable to findings investigating endogenous clearance here. In our studies, TCEP treatment normalized mucus, improved clearance, and reversed acute airway plugging (see Figs. 2–4). Thus, even under diseased conditions, mucolytic effects were examined within physiologic constraints.

Our findings suggest that an inhaled mucolytic treatment could confer acute protection from obstruction in asthma. Airway narrowing initiated by smooth muscle contraction is clearly important, and obstruction is amplified by mucus[15]. In bronchoalveolar lavage (BAL) fluid from patients with mild-to-moderate asthma, TCEP rapidly depolymerizes MUC5AC and MUC5B (Supplementary Fig. 7), demonstrating the presence of targets in asthma patients even under stable disease conditions. To translate our findings in mice to a clinically relevant setting, it will be crucial to achieve pharmacokinetic and pharmacodynamic indices that are effective, tolerable, and safe.

Full reduction of disulfide bonds could result in disruption of mucins to monomer fragments that form a poorly transported viscous liquid. In our studies, partial depolymerization of mucins (Fig. 2d, e and Supplementary Fig. 5) was sufficient to facilitate mucociliary clearance such that eosinophil numbers decreased by 81% in vivo (Fig. 2f). Accordingly, partial reduction could be effective for destabilizing aggregated mucus in patients with heterogenous muco-obstruction in large and small diameter airways. Like other inhalation therapies, the ability to achieve distal lung deposition will also depend on aerosol droplet sizes and chemical composition.

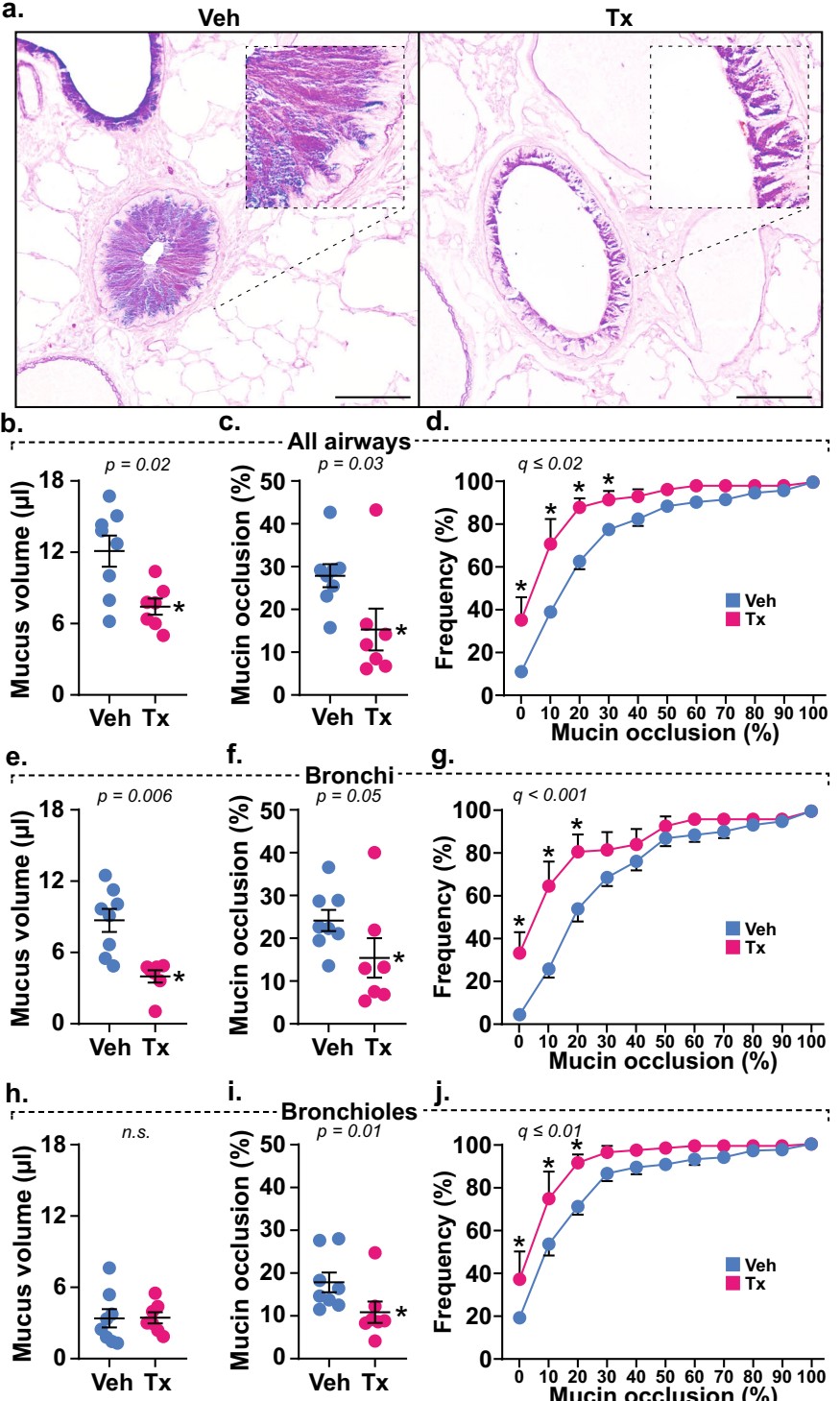

**Fig. 4 Mucolytic treatment disrupts mucus plugging. a** AB-PAS stained lungs obtained after methacholine dose response tests show mucin glycoproteins obstructing airspaces in vehicle (Veh, *n* = 8 biological replicates) treated mice that are disrupted by mucolytic treatment (Tx, *n* = 7 biological replicates). Scale bars, 500 µm (low power) and 25 µm (high power). **b–j** Calculated mucus volumes (**b**, **e**, **h**), mean fractional occlusion (**c**, **f**, **i**), and heterogeneous plugging (**d**, **g**, **j**) were significantly decreased in AOE-challenged mice receiving mucolytic (Tx, magenta, *n* = 7 biological replicates) compared to controls (Veh, cyan, *n* = 8 biological replicates). Mucolytic treatment significantly reduced obstruction across all airways (**b–d**), and this effect was most prevalent in bronchi (**e–g**). Although total mucus volume in bronchioles was low (**h**), occasional mucus aggregates were obstructive and are sensitive to TCEP treatment (**i**, **j**). Data in **b**, **c**, **e**, **f**, **h**, and **i** are means ± sem on scatter plots, with *p*-values shown and significance (*) indicating *p* < 0.05 by two-tailed Mann–Whitney *U*-test. Cumulative frequency distributions in **d**, **g**, and **h** show percentages of airways demonstrating occlusion, with circles and error bars identifying means ± sem, and "*" demonstrating significance by *t*-test using a two-stage step-up method at a 5% false discovery rate. Source data are provided as a Source Data file.

Recent reports have shown that mucin concentration and polymerization are critical factors that determine mucus viscoelasticity[8,32]. A prolonged disulfide reduction strategy similar to what was applied here was shown to improve chronic inflammation, airway injury, and abnormal tissue repair in a mouse model of pulmonary fibrosis[13]. Furthermore, TCEP was able to disrupt plugs in chronically mucus obstructed airways in a mouse model of CF[21].

The findings reported here build upon these prior works by showing mucolytic efficacy in a common disease, by demonstrating protective mucolytic effects during acute hypersecretion, and by validating that reversing mucus plugging has direct effects on pulmonary function (see Figs. 1 and 4). Taken together, these results support the concept that protection could be conferred in both chronic and acute respiratory diseases by preventing or reversing mucus dysfunction by disrupting mucin disulfide bonds.

Nonetheless, in addition to disulfide targets formed during mucin biosynthesis, there are other potential targets for mucolytic intervention. These include non-disulfide covalent linkages such as sugars added during glycosylation, as well as non-covalent mucin polymer interactions formed during mucin packaging into secretory granules. In addition, in the post-secretory environment oxidant-mediated mucin cross-linking is observed in cystic fibrosis (CF)[33] and asthma[2], and it is reported to occur on free thiols in these in static mucus aggregates in these settings.

Additional non-covalent mucus interactions associated with macromolecules such as DNA[34], or effects mediated by ionic and pH environments may have additional effects on interstitial fluid surrounding mucus mesh networks and could be also targeted[35–37]. Although these are not directly affected by reduction per se, agents that disassemble disulfides could be applied in combination with other therapies to improve mucus hydration and transport. Thus, treatments that reverse mucus dysfunction could then be considered as possible adjunct therapies.

Having identified the ability to deliver a mucolytic agent, promote mucociliary clearance, and reverse AHR in a mouse model of allergic asthma, findings reported here support the concept that mucin disulfide disruption could be applied to human lung pathologies where mucus dysfunction is significant[7,8]. Inflammatory or injurious causes of mucus dysfunction vary within and across diseases, yet they still result in a similar outcome—excessive polymeric mucin production. Indeed, mucus from fatal asthma exhibited comparable microstructure to CF sputum, and both samples had almost identical concentrations of mucus solids (Supplementary Fig. 8) despite having unrelated causes for abnormal hydration.

To this end, even in diseases with diverse primary causes, there could be benefits that derive from optimizing airway mucus. Furthermore, while reducing obstruction, mucin depolymerization could enhance the deposition and absorption of inhaled bronchodilator, anti-inflammatory, or antibiotic agents. Additional studies focused on developing effective mucolytic therapies and potentially mucin-selective treatments are needed.

## Methods

**Human specimens**. BAL fluid was obtained from volunteers with asthma recruited for a University of Colorado Institutional Review Board (IRB) approved study (ages 18–53, $n = 2$ male, 2 female, non-smokers). Volunteers gave informed consent in accordance with the Declaration of Helsinki. BAL was collected from the left upper lobe (lingula projection), and a 2 ml sample was separated and stored "neat" (without centrifugation) to preserve high molecular weight mucins that sediment at low centrifugation speeds.

For fatal asthma samples, samples were obtained from lungs donated to Oregon Health and Sciences University (kindly provided by Dr. David Jacoby) and National Jewish Health. Samples from both subjects were de-identified prior to use and were thus human subject exempt. Tissue sections were made from formalin-fixed paraffin embedded samples from both donors. Mucus aspirated from the trachea and lobar bronchi of samples at National Jewish health.

Spontaneously expectorated CF sputum samples were collected from patients (ages 31–52, $n = 3$ male/2 female, non-smokers, no CFTR modulator therapies) at the adult CF clinic at Johns Hopkins University. Studies were approved by the Johns Hopkins IRB, and participants gave informed consent as above.

**Mucin protein detection**. Histochemical staining with alcian blue-periodic acid Schiff's (AB-PAS) stain was performed using standard techniques[38]. For immuno-detection, human, and mouse mucins were detected using rabbit-anti-human MUC5AC (MAN5AC, Dr. Thornton's laboratory, diluted 1:1000)[39], mouse-anti-MUC5AC (clone 45M1, ThermoFisher, diluted 1:1000), rabbit-anti-human MUC5B (H300, Santa Cruz, diluted 1:5000), and rabbit-anti-mouse Muc5b (Dr. Evans's laboratory, diluted 1:5000)[1]. For biochemical labeling, mucins were also detected using biotinylated *Ulex europaeus* agglutinin I (UEA-I) lectin to detect fucose residues (Vector, Burlingame, CA) or with biotinylated maleimide to alkylate and detect reduced sulfhydryls (ThermoFisher).

**Immuno-/ lectin blotting**. Lung lavage and mucus specimens were separated on SDS/agarose gels to determine mucin profiles and changes in mucin polymer sizes upon reduction[40]. A dot-blot ELISA was performed to assess MUC5B levels in each human lavage sample, and lanes were equilibrated to contents of MUC5B, the predominant mucin in these patients with non-exacerbated asthma. In allergic mice, although lung lavage fluid contains mixtures of both Muc5ac and Muc5b, they are found at higher concentrations and can be more homogeneously sampled (see below). Therefore, mouse lung lavage samples were loaded into gels using equal volumes. After electrophoresis and transfer to PVDF membranes by vacuum blot[40], mucins were probed with antimucin antibodies (1:1000–5000 dilutions as indicated), and then detected with goat antimouse IRDye 680RD or goat antirabbit IRDye 800CW antibodies (LI-COR, Lincoln, NE, diluted 1:20,000). Image analysis was performed using Image Studio software (LI-COR).

**Particle tracking in human mucus**. For human airway mucus samples, 2.0 μm fluorescent carboxylated microspheres (ThermoFisher) were used to estimate mucus microrheology. Frozen mucus samples from the patients above were thawed on ice, distributed into 100 μl aliquots. A 10 μl solution of TCEP (100 mM) or saline (vehicle) containing 100,000 microspheres was added to each mucus sample. Samples were incubated at 37 °C for 10 min.

Immediately after incubation, samples were loaded onto chambered slides and video imaged at five randomly chosen sites on a BX63 microscope for 30 s (7.5 frames per sec) using a DP80 camera (Olympus, Center Valley, PA). Particle tracking was performed using Olympus cellSens software, and analyses were made to derive MSD values using a Matlab program (Mathworks, Natick, MA) that was published previously[25,26,41]. Using the frequency-dependent Stokes–Einstein equation, we translated MSD values to obtain approximations of micro-rheological properties of airway mucus[22,23].

**Mice**. Studies were conducted with approval of the University of Colorado Denver and Johns Hopkins University Institutional Animal Care and Use Committees. Housing rooms are maintained at 22 °C, 30–40% humidity, and a 14/10 (h/h) light/dark cycle and at least 12 fresh-air changes per h. Male and female BALB/cJ wild type mice were purchased from the Jackson Labs (Bar Harbor, ME). $Muc5ac^{-/-}$ mice were previously crossed onto a congenic BALB/cJ strain background[15]. Animals were housed under specific pathogen-free conditions and used in allergic asthma studies beginning at ages 6–8 weeks.

To induce allergic inflammation and mucin overproduction, mice were challenged using aerosolized *Aspergillus oryzae* extract[15] (AOE; Sigma), as shown in Supplementary Fig. 2. Mice received four weekly challenges, and endpoint analyses were studied 48 h after the last AOE challenge. To induce mucolysis, mice and mucus samples were exposed to tris(2-carboxyethyl)phosphine (TCEP, neutral pH solution, Thermo Scientific, Cat. no. 77720). Mice exposed to saline challenges were used as controls.

**Particle tracking in sputum samples and freshly excised mouse tracheas**. MIPs were prepared using 5-kDa methoxy-polyethylene glycol (PEG)-amine (Creative PEGWorks, Chapel Hill, NC) molecules that were densely conjugated to the surface of 100 nm carboxyl-functionalized polystyrene beads (PS-COOH; Product #F8801, 580/605, Molecular Probes, Eugene, OR)[25]. Covalent attachment to the carboxyl end groups on the PS-COOH particles was achieved using a crosslinker, 1-ethyl-3-(3-dimethylaminopropyl) carbodiimide hydrochloride (Sigma-Aldrich, No. E7750) and sulfo-N-hydroxysulfosuccinimide sodium salt (Sigma-Aldrich, No. 56485) in borate buffer (200 mM, pH 8.0, Growcells, No. MRGL-1100). Physicochemical properties of MIPs were then measured by a Zetasizer Nano ZS90 (Malvern Analytical, Malvern, UK). Resulting 100 nm MIPs exhibited hydrodynamic diameters of $98.0 \pm 6.9$ nm, polydispersity indices of $0.1 \pm 0.01$ and $\zeta$-potentials (i.e., indicative of particle surface charges) of $-4.1 \pm 0.2$ mV measured in 10 mM NaCl pH 7.4 at 25 °C.

CF and asthmatic mucus samples were stored at 4°C immediately after collection for up to 24 h for particle tracking experiments to ensure that the inherent mucus microstructure is preserved[42]. Aliquots of 30 μl of sputum were added to custom microscopy chambers. Next, 0.5 μl of MIP (0.00001% v/v) were

mixed gently with sputum sample to evenly distribute particles within the sample. The chamber was sealed with a circular coverslip and incubated at room temperature for 30 min prior to imaging.

Fluorescent MIPs were administered to the mucosal surfaces of tracheas freshly dissected from mice. Briefly, tracheal tissues harvested from animals were cut along the dorsal cranial-caudal axis and laid flat to expose mucosal surfaces. MIPs were diluted in saline (i.e., vehicle control) or TCEP (500 mM) at 0.02% w/v and administered in <1 µl volumes onto mucosal surfaces of the flat-mounted tracheas using a vibrating mesh nebulizer (Aerogen Solo, Chicago, IL) controlled by an Analog Discovery 2 data acquisition device (Digilent, Pullman, WA). After treatment with MIPs in vehicle or TCEP, tracheas were laid face down on a coverslip-bottomed dish (Thermo Scientific) and sealed with vacuum grease and parafilm, followed by incubation for 30 min at 4°C to temporarily immobilize cilia prior to particle tracking experiments[43].

Motions of MIPs were captured at 15 frames per second (i.e., exposure time of 67 ms) for 20 s on an Axiovert D1 inverted epifluorescence microscope (Zeiss, Chicago, IL) equipped with a Photometrics Evolve EMCCD 512 camera (Photometrics, Tucson, AZ) and a MetaMorph software (Molecular Devices, San Jose, CA). MSD values are averages of squared distances traveled by individual particles at a given time interval (i.e., timescale) and thus are directly proportional to particle diffusion rates. Tracking resolution was determined by analyzing MSD on 100 nm MIP particles immobilized in glue between two coverslips, which was found to be less than $\log_{10}(MSD_{\tau\,=\,1s})$ of −3.

**Acute endogenous clearance and mucolytic testing**. AOE challenged mice were exposed to TCEP (5 mM, 50 mM, or 500 mM) or saline aerosol for 40 min, at the end of which they were immediately used in lung lavage studies. Animals were anesthetized with urethane (2 g kg$^{-1}$, i.p.) and then tracheostomized with a 20 gauge blunt tip Luer stub[40]. Within 5 min of removal from aerosol treatment, lavage was performed using saline to isolate mucins and inflammatory cells. Acute endogenous clearance (AEC)[13] was determined by quantifying an acute decrease in leukocyte numbers in lavage fluid in TCEP relative to saline treated animals.

Lavage macrophages, lymphocytes, neutrophils, and eosinophils were enumerated as described above for human BAL. Remaining lavage fluid was divided into two aliquots. One portion was centrifuged at 1200 rpm for 10 min at 4 °C for subsequent studies of cytokines and other soluble factors. The other portion was treated with 1 M iodoacetamide (1/100 volume) to quench drug activity and alkylate thiols liberated by disulfide reduction; this portion was stored neat. Both were frozen on dry ice and stored at −80 °C.

**Airway hyperreactivity measurements**. Mice anesthetized with urethane (2 g kg$^{-1}$, i.p.), tracheostomized with a beveled blunt Luer stub, placed on a flexiVent (Scireq, Montreal, Quebec). Ventilated mice were paralyzed with an initial 0.2 ml injection of succinylcholine chloride (300–35 mg kg$^{-1}$, i.p.) to prevent spontaneous respiration. Mice were then cannulated with a 20 gauge catheter in the inferior vena cava. To maintain paralysis throughout the experiment, a continuous infusion of succinylcholine was administered as described below.

Total lung resistance ($R_L$), airway resistance ($R_{AW}$), tissue resistance ($G_{TI}$), and tissue elastance ($H_{TI}$) were measured at baseline and in response to methacholine (MCh), which induces bronchoconstriction and mucin secretion[15]. MCh was diluted to a final concentration of 1 µg MCh g$^{-1}$ body weight ml$^{-1}$ of normal saline solution also containing succinylcholine chloride (20 µg ml$^{-1}$ final concentration). This was administered using a digital infusion pump with rates adjusted to 4, 8, 16, 32, and 64 µl min$^{-1}$. MCh doses were given for 4–5 min with recordings of lung function made 20 times during this period. Between doses of MCh, TCEP (500 mM) was administered through an in-line ultrasonic nebulizer (10 s per dose, ~1–2 µl delivery to the lungs per dose). After the last dose of MCh was delivered, lungs were either lavaged with saline to evaluate mucin depolymerization by immunoblot, or fixed to assess mucus plugging[40].

**Histology**. To assess plugging, lungs were fixed transmurally at end-expiratory volume during responses to the highest dose of MCh used. Lungs were fixed in methacarn for 24 h, then transferred to absolute methanol. Fixed lungs were excised after 30–60 min and placed in a scintillation vial filled with methacarn. Lung volume was calculated using volume displacement in absolute methanol. Lungs were then cut into 2 mm cubes (~30 per lung), paraffin embedded in random orientations, and sectioned[15,40]. AB-PAS stained tissues were then scanned on an Olympus BX63 microscope, and each tissue fragment was imaged from the center using a 10× objective. Total airway vs. parenchyma volume fractions were determined using a 250 × 250 µm grid, and mucin volume fractions were determined using a 50 × 50 µm grid. Volume fractions were then normalized to lung and airway volumes.

**Statistical analysis**. Statistical analysis and graphs were performed in Prism (GraphPad, San Diego, CA). Comparisons were made using *t*-tests, Mann–Whitney *U*-tests, or ANOVA with post-hoc analyses as noted.

**Reporting summary**. Further information on research design is available in the Nature Research Reporting Summary linked to this article.

## Data availability

All data supporting the findings of this study will be available from the corresponding authors upon reasonable request. Source data are provided with this paper.

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

## Acknowledgements

This study was funded by NIH grants HL080396 (C.E. and C.M.) and ES023384 (C.E.), HL130938 (C.E. and W.J.), HL125169 (J.H.); by U.S. Department of Defense grants W81XWH-17-1-0597 (C.E.), W81XWH-19-1-0172 (C.E.), and PR192068 (C.M.); Cystic Fibrosis Foundation grants EVANS18IO (C.E.), JARAMI20F0 (A.J.), and HANES16XX0 (J.H.); the National Science Foundation NSF CAREER 1941401 (C.M.), and by the Medical Research Council grant MR/R002800/1 (D.T.).

## Author contributions

L.M., S.S., C.M., J.H., J.S., D.T., F.H., W.J., W.T., and C.E. designed the study; L.M., A. J., S.S., D.R., N.E., V.R., N.H., A.H., J.N., C.H., H.E., D.V., and G.D. planned, performed, and analyzed experiments. L.M. and C.E. wrote the manuscript with help from all coauthors. C.E. supervised the manuscript preparation.

## Competing interests

The muco-inert particle technology described in this article is being developed by Kala Pharmaceuticals. J.H. declares a financial, a management/advisor, and a paid consulting relationship with Kala Pharmaceuticals. J.H. is a cofounder of Kala Pharmaceuticals and owns company stock, which is subject to certain restrictions under Johns Hopkins University policy. C.E. is a paid consultant with Eleven P15, a company focused on early detection and treatment of pulmonary fibrosis. C.M. is a paid consultant with Sharklet Technologies, a company that uses surface texture to reduce biological adhesion to medical devices. The terms of these arrangements are being managed by Johns Hopkins University and the University of Colorado in accordance with respective institutional conflict-of-interest policies. W.T. and D.V. are employees of Parion Sciences, Inc., a company that designs and tests novel mucolytic agents. No proprietary mucolytic agents were used in this study. All other authors declare no conflicts of interest.
