## [Peer Review File · Nature Communications]

Reviewers' comments:

Reviewer #1 (Remarks to the Author):

With the exception of dornase alfa, used for the treatment of cystic fibrosis lung disease, the use of mucolytic drugs for the therapy of pulmonary diseases remains controversial. Most clinical studies have shown that existing thiol mucolytics provide little or no benefit.

These investigators have meticulously evaluated the effect of TCEP, as a rapidly acting reducing agent, on the in vitro structure of mucus plugs from subjects with severe or fatal asthma; as well as in a murine model of fungal induced allergy and asthma. They demonstrate that in human asthma mucus, the pore size, as evaluated by the multiple particle tracking (MTP) technique, is tighter leading to exclusion of the probes and that this can be reversed using TCEP. They also show that TCEP improves mucociliary clearance (MCC) and abolishes airway hyper responsiveness to methacholine in the allergic mice. This parallels the protection seen in Muc5ac knockout mice. The manuscript is well written and experiments appear to be meticulously conducted by an outstanding group.

I have the following questions for the authors.

1. For MTP, the probes (particles) are coated to lower surface tension, and transport following TCEP suggested an enlarging of pore size. It would be useful to know if this change in pore size is uniform or if there is a selective effect. Presumably, this can be done using image analysis. It was also assumed that MTP was uniquely determined by pore size. Is it possible that transport is also retarded by the fluid within and surrounding the pores and that the rheology of this fluid changes with TCEP exposure?

2. It is suggested that the mucolytic effect of TCEP was responsible for decreased airways hyperresponsiveness and inflammatory cells in bronchoalveolar lavage fluid. As a reducing agent, TCEP has antioxidant properties and this could decrease AHR and cellular inflammation as well.

3. The potential clinical utility of TCEP or similar drugs may be overstated, as this may be limited by

[1] Over liquification of secretions, as this will impair mucus transport. Newtonian liquids are poorly transported by cilia. It is thus important to determine the concentration dependent effects of a mucolytic on MCC.

[2] Inability of the drug to penetrate to the deepest airways in patients with pulmonary disease. Persons with fatal asthma have obstruction of airways down to the acinus.

Bruce K Rubin

Reviewer #2 (Remarks to the Author):

The research paper by Morgan et al. refers to the development of a novel asthma therapy that targets obstruction of airflow by hypersecreted mucins using inhalation of a fast-acting reducing agent with efficacious mucolytic activity. Authors demonstrate the depolymerization of mucin in vitro and the reversion of associated pathological effects ex vivo and in vivo in a mouse allergic model. On the molecular level their approach is based on the reduction of covalent disulfide bonds that link MUC5AC monomers via their vWF domains to macromolecular assemblies. There is no doubt that the content of this paper has high medical impact with respect to mucus obstruction in asthmatic patients.

On all levels of experimental approaches the study is convincing and the conclusions drawn by authors are fully supported by experimental evidence. No major objections to the experimental settings are made, at least concerning those parts that refer to mucin chemistry and biology. The theoretical background of all parts referring to mucin structure and physiology is sound. There remain only a couple of minor points that should be addressed during revision of the text (see below).

Minor points

Fig. 2: The figure legend needs revision, as the lectin blot in d. is referred to as under c. Also the total leukocytes recovered in lung lavage are actually shown in c., not d. etc.

The abbreviation MCh should be defined or explained to readers, when it is used first on page 6, last paragraph. In this context I would suggest to reduce in general the use of abbreviations in this paper, as it is full with "MCC, MPT, MSD, MIP, and MCh". Less involved readers may get lost and spelling out at least some of these abbreviations would make the text more readable.

In the experimental section I could not find information on the reagent TCEP, neither its chemical designation nor its source.

Page numbering is lacking in the provided pdf file.

Reviewer #1 (Remarks to the Author):

With the exception of dornase alfa, used for the treatment of cystic fibrosis lung disease, the use of mucolytic drugs for the therapy of pulmonary diseases remains controversial. Most clinical studies have shown that existing thiol mucolytics provide little or no benefit.

These investigators have meticulously evaluated the effect of TCEP, as a rapidly acting reducing agent, on the in vitro structure of mucus plugs from subjects with severe or fatal asthma; as well as in a murine model of fungal induced allergy and asthma. They demonstrate that in human asthma mucus, the pore size, as evaluated by the multiple particle tracking (MTP) technique, is tighter leading to exclusion of the probes and that this can be reversed using TCEP. They also show that TCEP improves mucociliary clearance (MCC) and abolishes airway hyper responsiveness to methacholine in the allergic mice. This parallels the protection seen in Muc5ac knockout mice. The manuscript is well written and experiments appear to be meticulously conducted by an outstanding group.

I have the following questions for the authors.

We thank Reviewer 1 for his careful and encouraging review. We have addressed concerns that were raised below.

1. For MTP, the probes (particles) are coated to lower surface tension, and transport following TCEP suggested an enlarging of pore size. It would be useful to know if this change in pore size is uniform or if there is a selective effect. Presumably, this can be done using image analysis. It was also assumed that MTP was uniquely determined by pore size. Is it possible that transport is also retarded by the fluid within and surrounding the pores and that the rheology of this fluid changes with TCEP exposure?

We thank the reviewer for raising this point. It has been previously reported that pore sizes of airway mucus is intrinsically variable, and the heterogeneity is likely further elevated in muco-obstructive lung diseases (e.g. asthma, CF, COPD) as expected from the highly complex nature of their pathogenic processes. Heterogeneity of mucus is an important issue. Indeed, the revised version of extended data Fig. 3a demonstrates that the median +/- interquartile range of MSD values measured in mouse tracheal mucus is wider in allergic (AOE + Veh, cyan) compared to healthy (grey) mice where the range identifies variability or the spread of nonparametric data. However, we found that the range became narrower upon TCEP treatment (AOE + Tx, magenta) to the level comparable to that of healthy mice. This finding suggests that the impact of TCEP on pore sizes is likely uniform throughout the mouse tracheal mucus.

Regarding the bulk fluid properties, this is also a good point raised by the reviewer. Particle diffusion in airway mucus is affected by several factors, including adhesiveness and pore sizes of mucus network and interstitial fluid (i.e. fluid between pores) viscosity. Given that MIPs used in this study are muco-inert (i.e. non-mucoadhesive), we could exclude the impact of adhesive interactions. We then estimated pore sizes using the MSD values assuming that the viscosity of the interstitial fluid would not be significantly altered by TCEP treatment. In a small number of sputum samples collected from CF patients, we added TCEP to a final concentration of 10 mM (identical condition to Fig. 1) or an equal volume of vehicle, and used centrifugation to harvested interstitial fluid

(supernatant). We then measured and compared MSD values of 100 nm MIPs in these interstitial fluids from vehicle- and TCEP-treated sputum samples and found that MSD values were virtually identical regardless of TCEP treatment. Thus, TCEP treatment did not significantly perturb interstitial fluid viscosity. We note that CF sputum samples were used in this study since we were unable to acquire additional fatal asthma mucus samples during the time window of this revision.

2. It is suggested that the mucolytic effect of TCEP was responsible for decreased airways hyperresponsiveness and inflammatory cells in bronchoalveolar lavage fluid. As a reducing agent, TCEP has antioxidant properties and this could decrease AHR and cellular inflammation as well.

The reviewer raises a good point here, which could be significant in a long-term treatment regimen. In this case, because the studies are conducted in acute settings (~20 min for AHR, and 30 min for clearance studies), we don't anticipate an anti-inflammatory effect of reduction/anti-oxidant effects. This is now addressed in the manuscript (see p. 8, end of first full paragraph)

3. The potential clinical utility of TCEP or similar drugs may be overstated, as this may be limited by

[1] Over liquification of secretions, as this will impair mucus transport. Newtonian liquids are poorly transported by cilia. It is thus important to determine the concentration dependent effects of a mucolytic on MCC.

We thank the reviewer for raising this important topic. We have performed a dose response study to assess the relationships between partial-to-complete mucin reduction and mucociliary clearance (see revised Fig. 2e,f and Extended Data Fig. 5). In nose-only exposures, we report dose dependent changes in depolymerization and clearance.

[2] Inability of the drug to penetrate to the deepest airways in patients with pulmonary disease. Persons with fatal asthma have obstruction of airways down to the acinus.

We thank the reviewer for this comment, and we agree that there are certain caveats that can be better made. In the revised version of this manuscript, statements are better qualified. These are discussed on pp 8-9.

Reviewer #2 (Remarks to the Author):

The research paper by Morgan et al. refers to the development of a novel asthma therapy that targets obstruction of airflow by hypersecreted mucins using inhalation of a fast-acting reducing agent with efficacious mucolytic activity. Authors demonstrate the depolymerization of mucin in vitro and the reversion of associated pathological effects ex vivo and in vivo in a mouse allergic model. On the molecular level their approach is based on the reduction of covalent disulfide bonds that link MUC5AC monomers via their vWF domains to macromolecular assemblies. There is no doubt that the content of this paper has high medical impact with respect to mucus obstruction in asthmatic patients.

On all levels of experimental approaches the study is convincing and the conclusions drawn by authors are fully supported by experimental evidence. No major objections to the experimental settings are made, at least concerning those parts that refer to mucin chemistry and biology. The theoretical background of all parts referring to mucin structure and physiology is sound. There remain only a couple of minor points that should be addressed during revision of the text (see below).

Minor points

Fig. 2: The figure legend needs revision, as the lectin blot in d. is referred to as under c. Also the total leukocytes recovered in lung lavage are actually shown in c., not d. etc.

We thank the reviewer for catching this mistake. It has been corrected.

The abbreviation MCh should be defined or explained to readers, when it is used first on page 6, last paragraph. In this context I would suggest to reduce in general the use of abbreviations in this paper, as it is full with "MCC, MPT, MSD, MIP, and MCh". Less involved readers may get lost and spelling out at least some of these abbreviations would make the text more readable.

In the body text, we have eliminated abbreviations for methacholine (MCh), multiple particle tracking (MPT), and mucociliary clearance (MCC). We hope this improves readability.

In the experimental section I could not find information on the reagent TCEP, neither its chemical designation nor its source.

This has been added to the materials section.

Page numbering is lacking in the provided pdf file.

We apologize for the inconvenience. It was assumed that the numbering would be incorporated with the complied file from the journal website. Page numbers have been added within the manuscript body.

REVIEWER COMMENTS

Reviewer #1 (Remarks to the Author):

I appreciate that the authors have been fully responsive to my comments. No additional comments or concerns

Reviewer #3 (Remarks to the Author):

The manuscript by Morgan et al describes the use of the phosphine based TCEP as a mucolytica, showing in both human samples and in mouse model of asthma. The data is well presented and with convincing outcome that shows a clear mechanism for the action of the agent in depolymerizing MUC5AC and MUC5B. With these results, I personally think that the aim of the paper is obvious, and the last two sentences in the abstract is not necessary, but rather an attempt of the authors to oversell their story.

Major points:

- The use of phosphines as mucolytica in pulmonary diseases has already been described (PMID: 30212240). It is appreciated that previous report was using a mice CF model of mice, rather than targeting asthma. However, the similarity with previous research should be acknowledged. The existence of prior work also questions the novelty of the presented manuscript. The authors should also acknowledge that phosphines has been used in mucinbiochemistry to efficiently generate mucin momomers from intestine and saliva (PMID: 12498206, PMID: 12498206)
- In Figure 1d, 2d,e and E4-7, the term polymerized and depolymerized is used quite vaguely. Do the authors have any idea of how the mucin monomers are migrating in their gels, in order to appreciate the size of monomers and polymers. For the moment its quite unclear where for instance the cleaved off c-terminal of Muc5B is migrating. It is also unclear what their antibodies against Muc5B and Muc5AC is targeting. Are they capable of detecting the mucin domain. If not, there may be additional species of MUC5B and MUC5AC mucins that should be included in the polymer/depolymer ratio.

Responses to Reviewer Comments

Reviewer #1 (Remarks to the Author):

I appreciate that the authors have been fully responsive to my comments. No additional comments or concerns

We thank the reviewer for taking the time to consider the work presented here.

Reviewer #3 (Remarks to the Author):

The manuscript by Morgan et al describes the use of the phosphine based TCEP as a mucolytica, showing in both human samples and in mouse model of asthma. The data is well presented and with convincing outcome that shows a clear mechanism for the action of the agent in depolymerizing MUC5AC and MUC5B. With these results, I personally think that the aim of the paper is obvious, and the last two sentences in the abstract is not necessary, but rather an attempt of the authors to oversell their story.

We thank the reviewer for providing perspective here. Over-selling was not the goal. Rather, a broad conclusion was laid out to help appeal to wide ranges of readers who may not be deeply engaged in the field. It is helpful to have additional eyes looking at the text to gauge this. The last two sentences of the abstract have been merged in order to point to future directions or translational applications.

Major points:

- The use of phosphines as mucolytica in pulmonary diseases has already been described (PMID: 30212240). It is appreciated that previous report was using a mice CF model of mice, rather than targeting asthma. However, the similarity with previous research should be acknowledged. The existence of prior work also questions the novelty of the presented manuscript. The authors should also acknowledge that phosphines has been used in mucinbiochemistry to efficiently generate mucin momomers from intestine and saliva (PMID: 12498206, PMID: 12498206)

We appreciate the reviewer's request for broader context, acknowledgement of prior work, and better identification of the novelty of studies proposed here. The prior work on P3001 by Ehre et al is discussed here (Discussion, paragraph 6), and we point out that the studies here go well-beyond that work, especially in terms of functional effects in mouse models. The in vivo work of Ehre et al demonstrated only a reduction in mucus plug volume in the CF mouse model driven by beta-ENaC sodium channel overexpression. Additional biophysical readouts were carried out in expectorated sputum samples. The novelty of our findings relates to protection from acute airflow obstruction in asthma and the opportunity apply mucolytic treatments in a disease that affects as much as 10% of the population.

With respect to phosphines, we have added the reference to PMID 12498206 and stated clearly that TCEP is a well-known reducing agent (Discussion, paragraph 6). PMID 30212240 does not disclose the mucolytic compound tested, referencing P3001, and does not indicate which class of compound was used (e.g. thiol or phosphine). Again, it is well accepted that reducing agents can depolymerize mucin

networks in vitro and in vivo. However, the novelty of the work presented here, is that targeting mucins with small molecule drugs has the potential to safely and effectively impact pulmonary outcomes in asthma that are clinically relevant to humans.

- In Figure 1d, 2d,e and E4-7, the term polymerized and depolymerized is used quite vaguely. Do the authors have any idea of how the mucin monomers are migrating in their gels, in order to appreciate the size of monomers and polymers. For the moment its quite unclear where for instance the cleaved off c-terminal of Muc5B is migrating. It is also unclear what their antibodies against Muc5B and Muc5AC is targeting. Are they capable of detecting the mucin domain. If not, there may be additional species of MUC5B and MUC5AC mucins that should be included in the polymer/depolymer ratio.

We thank the reviewer for raising this important question. We went back and forth between “reduced/unreduced” and “polymerized/depolymerized.” We chose the latter in part because the word “reduced” could be confused as referring to a decrease in concentration. Since we refer to MUC5AC/Muc5ac and MUC5B/Muc5b as “polymeric mucins,” we feel that the term is consistent.

In most experiments, we run gels with a marker that is run to the bottom and then off the agarose gel. As such, proteins that remain are greater than or equal to 250 kDa. In cases where we run extensively reduced samples, monomers can be seen that are significantly greater in mass than 250 kDa. Examples of these can be seen in the uncropped immunoblots shown in Supplementary Figure 9. Specifically in panel 9c, lanes with asterisks show mouse mucins that are reduced to monomers (reduced in 10 mM DTT for 30 min at 50 C). Note that three bands are present, which likely represent differences in glycoforms of Muc5ac and Muc5b. This is also evident in panel 9e where reduced bands are visible for MUC5AC. When sampling from lavage fluid, we don’t often see full monomers. We take this to mean that there is either partial reduction or that more extensively reduced mucins are eliminated faster by MCC.

The reviewer also raises a good point about the antibodies used and what we learn from the studies here. In Fig 1d, we note in the legend that for both antibodies used (H300 from Santa Cruz for MUC5B and 45M1 for MUC5AC), signals decrease with reduction. Hence, the monomers seen lanes with the highest concentrations of mucin are faint. We do not know of C-terminal cleavage of MUC5B as an explanation here, but we do recognize that this has been described for MUC2 and MUC5AC. That processing is believed to occur in during biosynthesis in the ER and Golgi (PMID: 16787389). We appreciate that this may help us to better understand functional differences between MUC5AC and MUC5B. However, that interesting line of questioning falls outside the scope of this study.

Lastly, in mouse samples shown in Figs 2d and Supplementary Figure 5, we used a lectin that binds to alpha 1,2-fucosylated carbohydrates in the glycosylated mucin domains of both Muc5ac and Muc5b. Similar degrees of electrophoretic migration, and differences observed between reduced/unreduced are seen with both probes.

REVIEWERS' COMMENTS

Reviewer #3 (Remarks to the Author):

I have taken part of the new version of this article. I still think that propriety information of the ability of phosphines to depolymerize mucin gels in-vivo diminishes the novelty of the presented work. I acknowledge that the model is different and address a pulmonary disease of less incidence. I also find it amusing that the authors claim that previous publication does not reveal the identity of the P-3001 compound. This is despite that the main author obviously has more information about the compound (<http://digital.auraria.edu/AA00006564/00001/pdf>), having spent the phd about the effect of P-3001 as mucolytica and its effect in asthma. In the thesis the compound is referred to as a phosphine containing substance.

Responses to Reviewer #3

We thank the reviewer for reinforcing the need for ensuring significance and clarity. A response to the remaining concern from this reviewer is below:

Reviewer #3: *I have taken part of the new version of this article. I still think that propriety information of the ability of phosphines to depolymerize mucin gels in-vivo diminishes the novelty of the presented work. I acknowledge that the model is different and address a pulmonary disease of less incidence. I also find it amusing that the authors claim that previous publication does not reveal the identity of the P-3001 compound. This is despite that the main author obviously has more information about the compound (<http://digital.auraria.edu/AA00006564/00001/pdf>), having spent the phd about the effect of P-3001 as mucolytica and its effect in asthma. In the thesis the compound is referred to as a phosphine containing substance.*

Response: We apologize for any lack of clarity with respect to novelty and sharing of proprietary information.

Asthma is a disease that affects 10% of the world population and is treated with bronchodilatory and anti-inflammatory agents. While mucus hypersecretion is recognized as an important component of disease, it is untreatable at present. Hence, there is a level of clinical significance that is central here as well as novelty in showing how mucin disulfide disruption is protective.

The issue of not revealing the identity of the P-3001 compound previously was linked to work being conducted separately. The Evans lab was also working on studies investigating effects of mucolytics in pulmonary fibrosis. Due to the nature of that research program, an NIH-funded cooperative agreement, we were blinded to the structures of all compounds being used.

Those studies included comparisons of “fast-acting” and “slow-acting” compounds. TCEP was included as a control compound for the fast-acting group, and it was called P-3001 to prevent bias. At that time, we were aware of the nature of phosphine or thiol derivative reducing agent groups for the purposes of material handling and safety.

Accordingly, the thesis document referenced by the Reviewer in the link above refers somewhat ambiguously to P-3001 as a phosphine compound because of the timing of Ms. Morgan’s dissertation. Her thesis was submitted in 2017, and our work on fibrosis was published until 2018 (*Nat Commun.* 2018 Dec 18;9(1):5363). As projects have moved forward, we have been unblinded and chosen to refer to P-3001 as TCEP for frankness and clarity.